

# Numerical Evaluation of magnetic absolute Measurements with arbitrary distributed DI-Fluxgate Theodolite Positions

Heinz-Peter Brunke  and Jürgen Matzka

 GeoForschungsZentrum Potsdam (GFZ), Telegrafenberg, 14473 Potsdam, Germany

*Correspondence to:* Brunke (brunke@gfz-potsdam.de)

**Abstract.**

At geomagnetic observatories so called absolute measurements are used to determine the calibration parameters of the main three-axis-magnetometer and their long term drift. This allows to get the vector of the geomagnetic field in an absolute geographic reference frame over long periods of time in order to study the secular variation of the Earth's magnetic field.

Absolute measurements of the magnetic declination $D$ and inclination $I$ are done by means of a nonmagnetic theodolite with a fluxgate sensor mounted on its telescope parallel to the optical axis. A fluxgate measures the magnetic field component along its sensor axis. The reading $S$ of this magnetometer vanishes if it points towards a direction perpendicular to the field. For absolute measurements standard measuring scheme using six to eight such theodolite positions are established routines in magnetic observatories. These standard DI schemes allow for a simple numeric evaluation and cancel out the influence of
instrument parameters like sensor offset and misalignment angles between fluxgate sensor and telescope.

We present a numerical method that allows to evaluate measurements of an arbitrary number (minimum 5 as there are 5 independent parameters) and of arbitrary theodolite positions and exploit it to this end. We implement an instrument model to calculate the fluxgate reading $S$ in dependence of field, instrument parameters and telescope direction (herein after referred to as theodolite position). Inserting actual measured values gives one nonlinear equation for each theodolite position. Eventually
this is resulting in an overdetermined system of nonlinear equations. This system is solved in the sense of a least square solution using the Gauss-Newton-method generalized to an overdetermined system. The accuracy of the resulting $D$, $I$ and base values is given in terms of estimated variances.

The accuracy of the resulting $D$ and $I$ values depends on both, the choice of used theodolite directions and on the accuracy of the measurements. The quality of each individual measurement can be assessed by means of calculated residuals.
A general approach has significant advantages. The method allows to seamlessly incorporate additional measurements for higher accuracy. Individual erroneous readings are identified and discarded without invalidating the entire data set. We show how a-priory information can be incorporated and how that allows to even evaluate a very reduced data set. We expect the general method to ease requirements for both manual and automated DI-flux measurements. It can reveal certain properties of the DI-theodolite which are not captured by the conventional method.
Based on the new method, a new measuring schema is presented. It avoids the need to calculate the magnetic meridian prior to the inclination measurements. Adjustment is always done with the same fine adjustment wheel, the one for the horizontal




circle. Leveling of the telescope is not necessary and thus leveling errors are avoided. All these makes the measurements faster and less prone to errors.

The option of using measurements off the normal DI positions makes measurements in the vicinity of the magnetic equator possible for theodolites without zenith ocular.

## 1  Introduction

The new method was motivated by a very practical reason. We wanted to enable measurements with a DI-flux theodolite without zenith ocular. Zenith oculars are very seldom available, but indispensable for making inclination measurements around the magnetic equator, where the magnetic field is horizontal. The need to evaluate DI-measurements at arbitrary positions was felt already before we started our work on a viable solution. Peter Crosthwaite (Crosthwaite, 1994) and Anna Willer (née Nilsson) (Nilsson, 2010) gave us useful personal notes on their approaches. Our work is partly based on their notes, but overcomes potential convergence problems faced in the previous work.

We took advantage of methods known from geophysical inversion theory like assessing the residuals, using a-priory information and objectively assessing the accuracy of the results. Exploitation of the method in practice showed benefits also for the routine work at observatories. For example, we show that at a given rate of erroneous readings, a higher percentage of successful absolute measurements can be achieved. We apply our method to various such data sets. We show the benefit for accuracy and reliability of routine absolute measurements and the resulting base values.

## 2  The conventional measuring schema for declination and inclination measurements

### 2.1  Practice of conventional D and I measurement

After leveling the theodolite, in the conventional D measurement the telescope is put to a horizontal position (reading of vertical circle: 90°). Then the two positions with the telescope pointing approximately to the East and the West are determined were the magnetometer reading is zero or small. This is repeated with the telescope flipped over (vertical reading:270°, sensor on the other side of the telescope). The four resulting readings of the horizontal circle ($D$ readings $\phi_{c1}...\phi_{c4}$) are used to calculate the direction of magnetic North (eq. 1). Subsequently the horizontal circle is adjusted to the magnetic North and the two positions on the vertical circle are determined, were the field reading vanishes. This is finally repeated with the horizontal circle adjusted to the magnetic South. These last four measurements $\xi_1...\xi_4$ serve to calculate the inclination $I$ (eq. 2). This conventional DI scheme hast two major advantages: It allows for a simple numeric evaluation just by average determination of respectively four readings. The influence of instrument parameters like sensor offset and misalignment angles between fluxgate sensor and telescope cancel out.

A detailed description of the measurement and the formulas can be found in (Jankowski and Sucksdorff, 1996; Matzka and Hansen, 2007; KringLauridsen, 1985; Kerridge, 1988).





In this paper we refer to $\phi$ as the geographic direction. $\phi$ is the actual reading of the horizontal circle of the theodolite corrected by the reading of the theodolite directed to a mark with known azimuth.

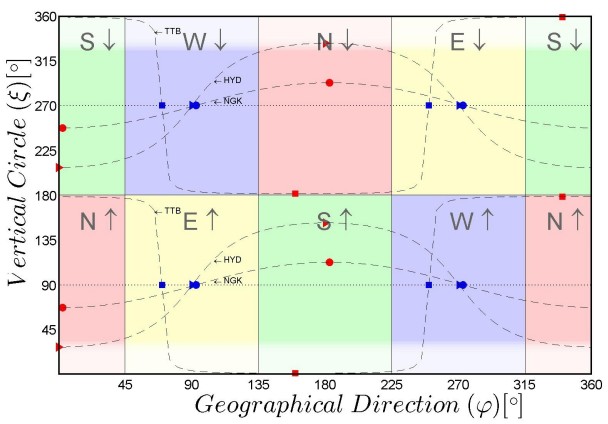
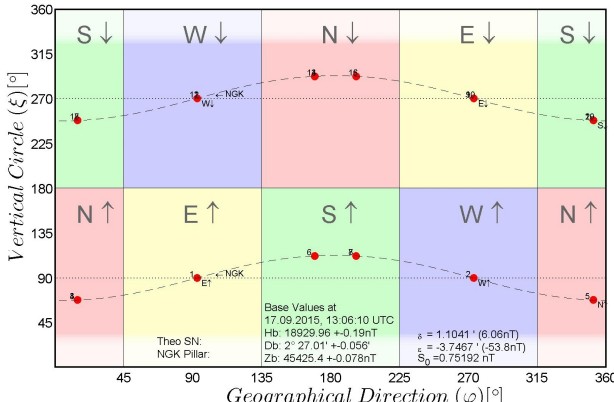

**Figure 1.** Graphic presentation of all possible telescope attitudes of a DI flux theodolite. The ordinate axis gives reading of the vertical circle of the theodolite. Values on the abscissa are readings of the horizontal circle corrected by an azimuth mark. The four colors code the cardinal direction of the heading of the telescope. The arrow indicates if the sensor is above or below the telescope.

**Left:** Dashed lines give positions with vanishing magnetic reading, specific to the three observatories Niemegk (NGK), Hyderbad (HYD) and Tatuoca (TTB). The positions possible at a given observatory are on or very close to the corresponding line. The marks represent $D$ (blue) an $I$ (red) measurements at these observatories according to the conventional scheme. Note the different $\phi$ values due to different local declination.

**Right:** Measurements taken at the Niemegk observatory. The scheme here is different to the conventional one, see text)

## 2.2 Evaluation of conventional D, I measurement

Calculating $D$ and $I$ for the conventional scheme means just calculating mean values, see (Jankowski and Sucksdorff, 1996)
5 as

$$
\begin{aligned}
D &= \frac{1}{4}(A_{W\uparrow} + A_{W\downarrow} + A_{E\uparrow} + A_{E\downarrow}) \quad \text{with:} \\
A_X &= \phi_i - \Delta D_i - \Delta A_i \quad ; \quad X \in \{W\uparrow, W\downarrow, E\uparrow, E\downarrow\} \\
&= \phi_i - \Delta D_i - \frac{360\circ}{2\pi}\frac{S_i}{H} \quad ; \quad i \in \{1,2,3,4\}
\end{aligned}
\tag{1}
$$

and, using $D$ to direct the telescope into the magnetic meridian plane, for the $I$ measurements

$$
\begin{aligned}
I &= \frac{1}{4}(V_{N\uparrow} + V_{S\downarrow} - V_{S\uparrow} - V_{S\downarrow}) + 90\circ \quad \text{with:} \\
V_X &= \xi_i - \Delta I_i - \Delta V_i \quad ; \quad X \in \{N\uparrow, S\downarrow, S\uparrow, N\downarrow\} \\
&= \xi_i - \Delta I_i - \frac{360\circ}{2\pi}\frac{S_i}{F} \quad ; \quad i \in \{5,6,7,8\}
\end{aligned}
\tag{2}
$$


$\Delta D_i$ and $\Delta I_i$ are corrections for field variations since the first measurement. $\Delta A_i$ and $\Delta V_i$ account for non zero magnetometer readings $S_i$.

Eight correct measurements are needed to apply these formulas. (Jankowski and Sucksdorff, 1996) give hints how certain individual errors can be identified. In practice though the entire measurement often is done over again, if just one single measurement is corrupted. For that reason it is good practice to do the entire measurement twice in the first place. We will show later how a more general method will make a much better use of the available measurements.

## 3 The general method for arbitrary orientated telescope positions

As explained above, the eight positions of the standard DI-scheme are used due to practical reasons. Theoretically only five measurements are sufficient to determine the five unknown quantities declination, inclination, two angles of misalignment and the magnetometer offset. There is no principal reason to do the measurements at exactly the positions as in the conventional scheme.

A general method allows to evaluate measurements at arbitrary theodolite positions. Also the number of measurements is arbitrary, but it must be at least five. If a priori information is used, even less are possible. The method has a lot of additional advantages. The quality of each single measurement can be assessed and outliers can be identified and discarded. The statistical accuracy of the resulting D, I values can be increased by additional measurements. Even though actual D, I measurements at the Niemegk observatory are known to be at a high accuracy level, using the new method we could still improve its accuracy and reliability. The quality of measurement could be assessed quantitatively.

### 3.1 Theory and numerical approach

#### 3.1.1 Instrument model

The DI-flux instrument model allows to calculate the magnetometer reading $S$ as a function of magnetic declination $D$ inclination $I$, total field strength $F$, theodolite readings $\phi$ and $\xi$ (horizontal circle and vertical circle) and the characteristics of the Sensor $\delta$, $\epsilon$ (misalignment angles in $\phi$ and $\xi$ direction) and the magnetometer offset $S_{off}$. Crosthwaite (Crosthwaite, 1994) gave a formulation based on vector geometry. The following formulation and its derivation based on spherical trigonometry can be found in (Nilsson, 2010):

$$
\begin{aligned}
S \quad &= c \cdot F \cdot \quad f(D, I, \delta, \epsilon, S_{off}, \phi, \xi) \\
&= c \cdot F \cdot \quad (-sin(I) \cdot cos(\phi + \epsilon) \\
&\qquad\quad + cos(I) \cdot sin(\phi + \epsilon) \cdot cos(D - \phi) \\
&\qquad\quad + cos(I) \cdot \quad \delta \quad \cdot sin(D - \phi)) + S_{off}
\end{aligned}
\tag{3}
$$

We used the latter because it facilitated the calculation of the partial derivatives for $D, I, \delta$ and $\epsilon$ that are needed for the Gauss-Newton method. The factor $c$ is the scale factor of the fluxgate magnetometer. It can be $c = 1$ or $c = -1$, depending on





the orientation of the sensor (1 for a positive reading $S$ if the telescope is pointing towards north.) Slight deviations of $|c|$ from 1 can be neglected, as long as S is small.

### 3.1.2 Accounting for field changes during the measurement by reduction to the first

Variations of the magnetic field since the time of the first measurement $t_0$ have to be taken into account for each later measure-

ment at time $t_i$. This is called reduction to the first measurement. Changes of $D$ and $I$ since the first measurement are small enough to justify the assumption of a linear instrument function:

$$
\begin{aligned}
S_i = \quad & c \cdot F \cdot f(D_0 + \Delta D_i, I_0 + \Delta I_i, \delta, \epsilon, \phi_i, \xi_i) + S_{off} & (4) \\
= \quad & c \cdot F \cdot f(D_0, I_0, \delta, \epsilon, \phi_i, \xi_i) + \frac{\partial f}{\partial D} \Delta D + \frac{\partial f}{\partial I} \Delta I + S_{off} & \\
& \text{with} \quad \Delta D = D_i - D_0 \quad \text{and} \quad \Delta I = I_i - I_0 & (5)
\end{aligned}
$$

### 3.1.3 Declination, Inclination and Instrument parameters by solution of a system of nonlinear conditional equations

Each measurement with the DI-Flux theodolite consists of measuring the angles $\phi$, $\xi$ and the magnetometer reading $S$. It delivers respectively one nonlinear equation for the unknown quantities $D$, $I$ and the instrument parameters $\delta$, $\epsilon$ and $S_{off}$. In the following the vector of unknowns to solve for, are comprised in the parameter vector $\mathbf{p} = (D, I, \delta, \epsilon, S_{off})$. For a total number of $N$ available measurements we get a system of $N$ equations:

$$c \cdot F \cdot f_1(\mathbf{p}, \phi, \xi) - S_1 \quad = \quad r_1 \tag{6}$$

$$...$$

$$c \cdot F \cdot f_N(\mathbf{p}, \phi, \xi) - S_N \quad = \quad r_N$$

with

$$S_i = S_{reading,i} - \frac{\partial F_i}{\partial D} \Delta D_i - \frac{\partial F_i}{\partial I} \Delta I_i$$

the magnetometer readings reduced to the first reading, and $\Delta D_i$, $\Delta I_i$ variations of $D$ and $I$ since the first reading. Ideally are $r_i = 0$. Generally we can expect that the number of equations exceed the number of unknowns. Hence the resulting nonlinear system is overdetermined. A solution of this system exists only in the sense of a least square solution minimizing $\sqrt{\sum r_i^2}$. Consequently additional conditional equations can be included seamlessly. Such additional equations can describe a-priory information. A-priori information stabilizes the solution, especially if the number of equations is small. Let the collimation

angles $\delta_{apriory}$ and $\epsilon_{apriory}$ be known from former measurements. These information can be accounted for using the following additional equations:

$$
\begin{aligned}
\frac{\sigma_S}{\sigma_{\delta_{apriory}}} (\boldsymbol{\delta} - \delta_{apriory}) &= r_{N+1} \quad \text{and} & (7) \\
\frac{\sigma_S}{\sigma_{\epsilon_{apriory}}} (\boldsymbol{\epsilon} - \epsilon_{apriory}) &= r_{N+2}
\end{aligned}
$$




The factors $\frac{\sigma_S}{\sigma_{\epsilon_{apriory}}}$ ans $\frac{\sigma_S}{\sigma_{\delta_{apriory}}}$ are indispensable to achieve the correct weighting of the a-priory information. The conventional deviation $\sigma_S$ of the magnetometer readings $S$ can be determined empirically investigating the distribution of the residuals $r_i$. The values $\sigma_{\delta_{apriory}}$ and $\sigma_{\epsilon_{apriory}}$ have to be guessed according to the precision of the given a-priori information.

### 3.1.4 A first guess of D and I

A good first guess is important for the Gauss-Newton method to converge towards the right solution. Instrument parameters $\delta$, $\epsilon$ and $S_{off}$ are small and can always be assumed to be zero as a first guess. Declination and inclination are achieved fitting a plane to all 3D pointing vectors $\mathbf{P}_i$ of the telescope neglecting the fluxgate readings $S_i$.

$$\mathbf{P}_i = \begin{pmatrix} cos(\phi_i)sin(\xi_i) \\ sin(\phi_i)sin(\xi_i) \\ -cos(\xi_i) \end{pmatrix} \quad ; \quad i = 1...N \tag{8}$$

The normal vector $\hat{\mathbf{n}}$ on the least square plan fit through this set of points $\mathbf{P}_i$ is used as a first, but already pretty accurate
approximation of the field direction. $D$ and $I$ are calculated accordingly. $I$ as the angle between $\hat{\mathbf{n}}$ and its projection to the horizontal plane and $D$ as the angle between the projection to the horizontal plane and the North direction just as in equation 11.

### 3.1.5 Solution of the nonlinear system with the Gauss Newton method

The Gauss Newton method is a straight forward extension of the well known Newton method from one to more dimensions.
It is also referred to as Newton-Raphson Method (Press et al., 1986). The method works by iteratively improving a first guess by a gradient method. The method requires to calculate the partial derivatives of the instrument model for all unknowns. It is known to converge quadratic, with is very fast. But convergence is only guaranteed, with an appropriate first guess.

For a given guess $\mathbf{p}$ the improvement $\Delta\mathbf{p}$ is found as solution of the linear system:

$$\mathbf{J}(p)\Delta\mathbf{p} - \mathbf{f}(\mathbf{p}) = \mathbf{r} \tag{9}$$

with the jacobien matrix $\mathbf{J}$. More explicitly:

$$\begin{pmatrix} \frac{\partial f_1}{\partial D} & \frac{\partial f_1}{\partial I} & \frac{\partial f_1}{\partial \delta} & \frac{\partial f_1}{\partial \epsilon} \\ & ... & & \\ \frac{\partial f_N}{\partial D} & \frac{\partial f_N}{\partial I} & \frac{\partial f_N}{\partial \delta} & \frac{\partial f_N}{\partial \epsilon} \end{pmatrix} \begin{pmatrix} \Delta D \\ \Delta I \\ \Delta \delta \\ \Delta \epsilon \end{pmatrix} - \begin{pmatrix} f_1(\mathbf{p}) \\ ... \\ f_N(\mathbf{p}) \end{pmatrix} = \begin{pmatrix} r_1 \\ ... \\ r_N \end{pmatrix} \quad \text{with N the number of measurements}$$

The functions $f_1(\mathbf{p})$ to $f_N(\mathbf{p})$ are just the instrument model $f_i(\mathbf{p}) = f(D, I, \delta, \epsilon, \phi_i, \xi_i)$ of equation 3 with the measured values $\phi_i$ and $\xi_i$ inserted.

In our case equation 9 is overdetermined and must be solved in the sense of minimizing $|\mathbf{r}|$. The according normal equations
read:

$$\Delta\mathbf{p} = \left(\mathbf{J}(p)^{\mathbf{T}}\mathbf{J}(p)\right)^{-1}\mathbf{J}(p)^{\mathbf{T}}\mathbf{f}(\mathbf{p}) \tag{10}$$





In equation 9 the instrument offset $S_{off}$ does not show up. For the sake of simplicity and numerical stability we separated its calculation from the other parameters. $S_{off}$ is just an additive term in all equations as shown in equation 3. Accordingly it can be calculated after each iteration just as the mean value of the residuals.

### 3.1.6 Baseline determination

At geomagnetic observatories the main three-axis-magnetometer always have a certain offset. One reason is, that the instrument dynamic range is often adapted to the natural variations of the field which can be far smaller than their constant part. Secondly, deviations between the sensor directions and the geographical reference frame need to be accounted for. Another reason is, that a slight drift due to mechanic and electronic instabilities can never be excluded. This is mainly due to geometric reasons. Already a very small rotational movement, e.g. in the instrument or its pillar can result in a measurable offset. Hence the three-axis-magnetometer often is referred to as variometer. Absolute measurements are needed to calibrate these variometers, i. e. determine the offsets and their long term drift. They are indispensable to determine the vector of the geomagnetic field in an absolute geographic reference frame over long periods of time.

The offsets are called "base values". The stability of the base values is indicative for the stability of the variometer, for the accuracy of the absolute measurements and for unwanted small scale local magnetic field changes. Thus they are a measure of the quality of an observatory.

Immediate results of absolute measurements are Declination and Inclination $D$ and $I$. Together with the total field intensity $F$ they are used to calculate the base values.

We assume that the variometer is set up with its X-sensor pointing towards the magnetic North, the Z-sensor downwards and consequently the Y-sensor towards East. Small misdirections are covered by the base values. Accordingly the measured field components $(X_{var}, Y_{var}, Z_{var})$ can be identified with (H, D, Z). We denote the respective variometer readings as $H_{var}$, $D_{var}$ and $Z_{var}$ and their offsets as $(H_B, D_B, Z_B)$. The following Eq. 11 shows that the Y-channel can be identified with variation of the angle $D$.

$$D_{var}[°] = arcsin(\frac{D_{var}[\text{nT}]}{H_{abs}}) \quad \text{or} \quad D_{var}[°] \cong \frac{360°}{2\pi \cdot H_{abs}} D_{var}[\text{nT}] \tag{11}$$

with

$$H_{abs} = F \cdot cos(I) \tag{12}$$

and finally

$$Z_B = Z_{abs} - Z_{var} \quad \text{with} \quad Z_{abs} = F \cdot sin(I) \tag{13}$$



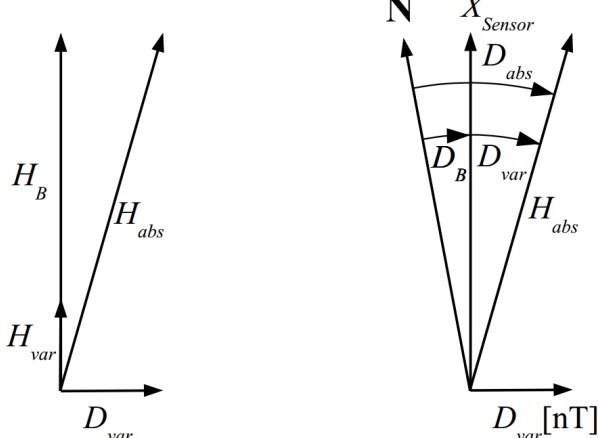

**Figure 2. (left)**: $D_{var}$ and $H_B + H_{var}$ form a rectangular square with $H_{abs}$ as hypotenuse. **(Right)**: $D_{abs} = D_B + D_{var}$

## 4  Results

### 4.1  Application of the method for outlier detection in traditional data sets

Outliers, which otherwise remained undetected could clearly be identified and discarded as shown in Figure 3a). In figure 3b) the according improvements of base values of H are flagged with red circles. The figure shows traditional H values in blue and

5   values calculated with the general method in red. Experience showed, that outlier identification became increasingly successful, the more measurements are taken. If only a restricted number of measurements is available it helps to use values of instrument parameters known from a time shortly before or after as a-priory information in order to successfully determine outliers.

### 4.1.1  Revelation of stray field effects

A good measurement is indicated by residuals ranging between about $-2nT$ to $2nT$ with a nice Gaussian distribution. This

10   results in an uncertainty of $B_H$ of about one half of a nT, as it is the case for major part of the measurements shown in Figure 3b. Figure 3a could give an example of a good measurement, if the outlier is discarded. Higher residuals give an indication to stray fields or systematic errors. In case of higher residuals is worthwhile to take a closer look at their distribution. Figure 4a shows residuals observed in two routine measurements at 5.Apr.2016 (each measurement 16 positions). Figure 3a shows that the residuals are clearly bigger than normal. We observed the same behavior on 12.Apr (3b). Our explanation is a problem in

15   magnetic cleanliness occurring in the first half of April 2016. Figure 4b shows a systematic error of a certain theodolite. The residuals go proportional to the cosine of twice the reading of the horizontal circle $\phi$. Reevaluating data taken one year before with the same theodolite showed exactly the same behavior (Zeiss Theo 020, 817992). We could produce a similar effect by sticking magnetically soft material to the alidade of a good theodolite.



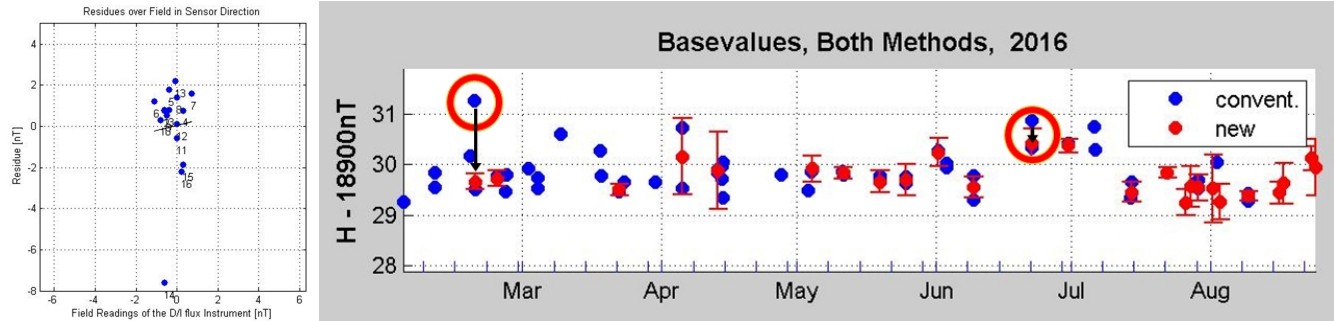

(a) Residuals of the measurements at 18.02.2016

(b) Basevalues of H calculated with conventional and new method

**Figure 3.** Panel(a) shows the identification of an outlier. Panal(b) shows in blue traditional base line data of NGK. Red points show data obtained with the new method. Red circles show baseline data which could be corrected. Calculated error bars allow to assess the quality of each measurement. The measurements taken in the first half of April are slightly disturbed, see figure 4a.

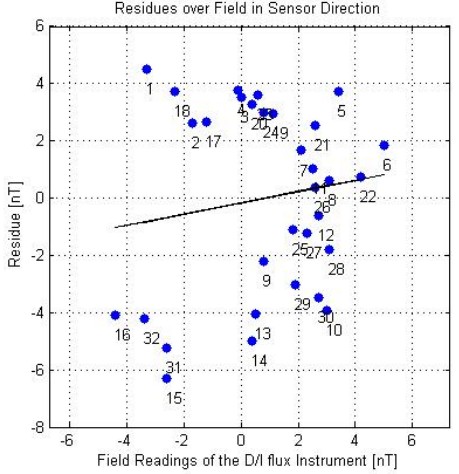

(a) Elevated residuals of the measurements at 05.04.2016

(b) Systematic dependance of residuals to the pointing direction of the adidade

**Figure 4.** Panel(a) shows unsystematically increased residuals, possibly by magnetic cleanliness problems during the measurements. Panel(b) shows residuals which have a sinusoidal dependance of the pointing direction of the alidade. Measurements with the sensor above the telescope in blue, else in red. The green dots show the differences to the fitted sin curve. This is could be due to a slightly magnetic instrument.





## 4.2  A new scheme for absolute measurements, easy, precise and less error prone

The new method allows to make D-I measurements with a simplified and less error prone scheme. Advantages with respect to the conventional one are:

- No need to calculate the magnetic meridian.

- The method tolerates imperfect leveling of the telescope.

- A measurement scheme that makes it easy to adjust the fine adjustment wheels is used throughout the entire measurement:

    1) Set vertical circle to a certain value.

    2) Adjust reading $S$ to almost zero with the horizontal circle and its fine adjustment wheel.

This scheme method begins with a repetition of the procedure know from $D$ measurement in the conventional D/I scheme. This is followed by further readings with the telescope tilted significantly off the horizontal plane. In the following we call it "**Adjust S horizontally**":

*For a given $\xi$ -value (vertical circle), adjust the magnetometer reading $S$ to 0 (or a small value) using the horizontal circle (coarse and fine adjustment). Note the reading of both circles, magnetometer reading $S$ and time.*

The right panel of figure 1 shows the distribution of data points for the new scheme.

Using the procedure defined above the **new scheme** reads (the first four positions identical to conventional D-measurements:

**1.** Set $\xi$ to 90° with the telescope pointing towards East. Adjust $S$ horizontally.

**2.** Invert the telescope, set $\xi$ to 270°, telescope to the West. Adjust $S$ horizontally.

**3.** Turn telescope to East, set $\xi$ to 270°, telescope to the East. Adjust $S$ horizontally.

**4.** Invert the telescope, set $\xi$ to 90°, telescope to the East. Adjust $S$ horizontally.

The next measurements are done with the telescope tilted by a certain angle $\alpha$ off the horizontal plane. Bothe upwards as well as downwards. The choice of $\alpha$ is not critical but it must be lower than $90 - I$, and not too small. At Niemegk is $I = 67.5°$ and $\alpha = 20°$ is good choice.

**5.** Set Set $\xi$ to $90 - \alpha$ with the telescope pointing towards North-North-East. Adjust $S$ horizontally.

**6.** Invert telescope, set $\xi$ to $270 - \alpha$ with the telescope pointing towards South-South-West adjust $S$ horizontally.

**6.a** (optional) Invert telescope, Set $\xi$ to $90 - \alpha$ with the telescope pointing towards North-North-West. Adjust $S$ horizontally.



**6.b** (optional) Invert telescope, set $\xi$ to $270 - \alpha$ with the telescope pointing towards South-South-East. Adjust $S$ horizontally.

7. Set Set $\xi$ to $90 + \alpha$ with the telescope pointing towards South-South-East. Adjust $S$ horizontally.

8. Invert telescope, set $\xi$ to $270 + \alpha$ with the telescope pointing towards North-North-West. Adjust $S$ horizontally.

**8.a** (optional) Invert telescope, Set $\xi$ to $90 + \alpha$ with the telescope pointing towards South-South-West. Adjust $S$ horizontally.

**8.b** (optional) Invert telescope, set $\xi$ to $270 + \alpha$ with the telescope pointing towards North-North-East. adjust $S$ horizontally.

**Remark:** In steps **1. ... 4.** there is no need to chose the $\xi$ values of 90 and 270 exactly. Values closed to these can also be used, but have than to be noted of course. It is a matter of personal preference if more effort is used to setting $\xi$ to a given value or to exactly read and note an arbitrary value. The latter is slightly more effort, but reading a value is more precise than setting it.

### 4.2.1 Avoiding problematic measuring positions measuring close to the magnet equator

At shallow geomagnetic inclination, within 2000 km to the North and South of the geomagnetic equator, the conventional DI-flux procedure involves vertical circle readings at steep telescope positions, which is not possible without zenith oculars mounted on the theodolite. This problem can be circumvented by using the new scheme just presented in the former section. Else than for the conventional scheme no measurement needs to be taken in the direction of the magnetic meridian. It is easy to avoid measuring positions too closed to the meridian. The tilt angel for the I measurements has to bee smaller than the inclination.

## 5 Discussion

We tested the method for more than one year in the Niemegk Observatory. A first test was of course to apply the method to data produced with the traditional scheme. In case of faultless data, we got exactly the same results, but this time with error bar. We also evaluated data that could not be treated traditionally because they were partly corrupted. The method still works fine if one or even two readings represent misreadings. A data set with the vertical reading set to 90Gon instead of 100Gon, produced by a person who was used to a degree scale, could be evaluated without problems. We found that a major advantage in observatory routine is, that several data sets measured at the same day can be evaluated as a single data set. This allows to improve the accuracy of the results due to statistics and to easily identify outliers. Investigating the calculated error bar allows to assess measuring conditions. As shown in Figure3(b) we could identify a problems in magnetic cleanliness in the first half of April. Figure 4(a) shows the respective residuals. Measuring in more than the four principal directions (see figure 4(b)) reveals



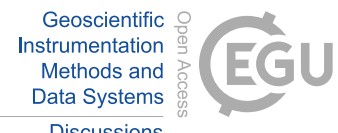
properties of the DI theodolite that otherwise remain hidden and gives a new perspective to assess the instrument. A simpler and less error prone measuring scheme was developed.



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
