# Peer review of "Numerical Evaluation of magnetic absolute Measurements with arbitrary distributed DI-Fluxgate Theodolite Positions"

_Geoscientific Instrumentation, Methods and Data Systems, 2017_

## Referee Comment (RC1) · Anonymous Referee #1 · 31 Mar 2017

**Revision of the manuscript "Numerical Evaluation of Magnetic Absolute Measurements with Arbitrary Distributed DI-fluxgate Theodolite Positions" (gi-2017-3).**

The manuscript "Numerical Evaluation of Magnetic Absolute Measurements with Arbitrary Distributed DI-fluxgate Theodolite Positions" deals with an alternative method to perform absolute measurements in geomagnetic observatories. This method is a more general one, and circumscribes the widely used 8-point scheme to a particular case of it. It takes advantage of the whole locus of perpendicular orientations of the DI-flux sensor with respect to the geomagnetic field vector, instead of restricting the observations to the meridian plane. An arbitrary number of readings corresponding to different theodolite orientations can then be introduced, which is meant to result in a higher accuracy of the target angles *D* and *I* by benefiting from a wider statistical population. The target angles are found by optimization in a least-squares sense by assuming a particular model of the instrument. Additional advantages of the method being argued by the authors are time saving and less error predisposition.

The manuscript might be useful to broaden the traditional scheme of observation, and especially for those low-latitude sites where it cannot be applied without specific (and perhaps difficult to find) and suitable (non-magnetic) theodolite accessories, but it lacks mathematical depth and conclusive arguments to support authors' theses.

From my point of view, the manuscript may be acceptable, but not before a substantial revision.

**Major concerns:**

- My main concern is that the presented model of the instrument is too simple: DI-flux theodolites, in general, are known to suffer from a number of drawbacks that make them imperfect instruments. The most popular ones are in fact the misalignment errors and the sensor offset, but there are others, like eccentricity of the graduated circles with respect to the rotation axes, non-orthogonality of these axes, or residual magnetizations (see, e.g., Marsal and Torta, 2007). Not accounting for them assumes that they are zero, resulting in biased values for the target angles *D* and *I*. From my point of view, the advantage of the conventional scheme is that the value of *D* is deduced from a set of 4 equations, while that of *I* is deduced from another set of 4 (virtually) *independent* equations. This allows, in practice, treating the misalignment/offset errors as independent in each set (even if we call them the same in both sets). This obviously does not guarantee that the *D* and *I* values deduced from the traditional method are the correct ones, but at least it allows for more degrees of freedom. I suggest the authors either extend their scheme to include these additional errors, or justify their simplification, but at least warn the reader that the presented method consists in a first approximation, not considering the whole complexity of real DI-flux instruments.

- If I'm not wrong, most equations have errors, showing a lack of rigor. Please, check them thoroughly:

- Eq. (1): I think the authors should subtract (or add) 180° in the r.h.s. For example, if D = 0°, one would obtain A_Eu = A_Wd = 90°, A_Ed = A_Wu = 270°, and the mean would be 180°.

- Eq. (2): a) The subscript of the last vertical reading should be N instead of S. b) The +90° serves only for the northern hemisphere; in the southern hemisphere a minus sign should be used.

- Eq. (3): Replace the occurrences of $\phi + \epsilon$ with $\xi + \epsilon$.

- Eq. (4): I think the factor $cF$ is lost in the terms of the partial derivatives. The authors should also say that the partial derivatives are to be evaluated in $D_0, I_0$.

- Last line of eq. (6): What is $F_i$ in the numerator of the partial derivatives? I think $cFf_i$ should be used instead.

- Eq. (9): I'm not sure this expression is correct. Indeed, the authors are not using the vector $S_i$ defined in eq. (6). I think the correct formula is $\mathbf{J(p)\Delta p} - (\mathbf{S} - \mathbf{f(p)}) = \mathbf{r}$.

- Eq. (10): Again, I think the correct formula is $\Delta \mathbf{p} = \left(\mathbf{J(p)^T J(p)}\right)^{-1} \mathbf{J(p)^T} (\mathbf{S} - \mathbf{f(p)})$.

Reference:

Marsal, S. and J.M. Torta (2007), An evaluation of the uncertainty associated with the measurement of the geomagnetic field with a D/I fluxgate theodolite, Measurement Science & Technology, 18, 2143-2156.

**Minor points:**

- English is not at the expected level. Please, revise it thoroughly. In the title, for example, I think the word "arbitrary" is an adjective rather than an adverb, so it should be replaced with "arbitrar**ily**". Also, punctuation is poor throughout the manuscript: commas, for example, are used where they should not, and are no longer used where they should. All these facts make it difficult reading through the manuscript.

- The title confused me the first time I read it. 'Position' for me can be understood as 'location', so I expected an article dealing with an ensemble of theodolites arbitrarily distributed. What about "reading positions", "orientations", "attitudes", "alignments", …? I would suggest the authors to ask a native English speaker.

- The abstract should be substantially reduced. Only information that is crucial for an overview of the article subject should be kept. Please, pass most of the abstract information to the Introduction and arrange it properly.

- Line 3: I think this not only applies to three-axis magnetometers, but also to others (like the dIdD coils) having 2 axes. What about using 'variometers' in general?

- Line 7: **nearly** parallel.

- Line 10: The use of the term "instrument parameters" sounds somewhat optimistic to me, especially when these parameters are said to be canceled out. Such "parameters" are usually

known as "errors" in the common language. As an intermediate solution, I suggest saying: "intrinsic errors (hereafter referred to as *instrument parameters*)".

- Line 11: The authors use the word "measurement" throughout, but I think they should define this term, as it is too generic in this case: is it a single orientation of the theodolite, or the whole set of them? I think the authors refer to *measurements* as *reading positions* or *alignments* of the theodolite.

- Line 12: Replace "theodolite positions" with "orientation".

- Line 17: Shouldn't it be 'D **and** I base values'?

- Line 27: I agree that the new procedure may be somewhat faster, but I guess not so much as the authors would like to suggest. They may save the time of computing the means for the magnetic meridian (1 minute or less for a skilled operator) and aligning the telescope exactly in the horizontal (meridian) position in the *D* (*I*) part, but in contrast, they must write down the angles corresponding to the vertical circle in each step of the *D* part (while they are known to be 90° 00' 00'' and 270° 00' 00'' in the conventional scheme), and the angles of the horizontal circle in each step of the *I* part (while they are known to be in the magnetic meridian in the traditional scheme). Can authors give a notion on how much faster is this procedure as compared to the conventional one?

- Line 28: As before, authors should give solid arguments to convince readers on the convenience of using this method. Please, compare the uncertainty that will be reached with this scheme to that of the traditional one.

- Combine sections 2.1 and 2.2 into one single section (so there are no sub-sections in Section 2). After that, try to arrange all the content in a more logical order. For example, the first paragraph is OK where it is; secondly, present the equations and define all the different variables; thirdly, put the current 2nd paragraph; then introduce Figure 1 in the text body (it should be clear what is it aimed at) and show it. The last paragraph in current section 2.2 is also OK where it is.

- First paragraph in section 2.1: Measurements are not necessarily done in this order: Eu, Wu, Ed, Wd, Nu, Sd, Nd, Su. If authors think this way of explaining allows being more precise, please say that this is an example of procedure, but that measurements can indeed be made in any order (provided that *D* measurements are made first).

- Lines 26-27 in section 2.1: to ease reading, mark the two referred advantages as a) and b).

- Eq. (1): the notation $i = 1 \ldots 4$ (as in eq. (8)) is normally used instead of $\in$. The same applies to the line above and eq. (2).

- Line 11 of section 2.2: Say that all the angles above are expressed in degrees.

- Line 13 of section 3.1.3: I think authors should put the subscript 0 in *D* and *I*.

- Eq. (7): Would it be suitable to introduce an equivalent additional equation for $S_{off}$?

- Line 20: Jacobi**a**n.

- In eq. (10), include an additional equation to calculate the error or uncertainty associated with the parameters in $\Delta \mathbf{p}$ according to the proposed method. I think the authors refer to

these error bars later in the caption of Figure 3 and in the Discussion, but the manuscript does not show how to calculate them.

- Line 5 of section 3.1.6: Again, why do the authors refer only to three-axis magnetometers?

- Line 20 in 3.1.6: I think it is a better practice to refer to $D$[nT] as the $E$ component. See, e.g., https://www.ngdc.noaa.gov/IAGA/vdat/IAGA2002/iaga2002format.html.

- Eq. (13): include an equivalent equation for $D_B$ (or rather $E_B$, see previous point).

- The Results section is presented before showing the details of the suggested procedure, which is not done until section 4.2 (almost at the end of the manuscript). This prevents the reader from judging the results, as he/she has not yet formed a concrete idea of the new scheme. I recommend placing the current sub-section 4.2 at least before 3.1.6.

- Line 6 of section 4, and line 26 in the Discussion: Could the method integrate measurements made in different days? If so, how many days can it integrate?

- Figure 3: It is not clear to me what is represented in the abscissa of Figures 3a and 4a. Is it $S_i$ of eq. (6)? Please, explain in the text body what is exactly represented in each figure, and summarize it in the figure captions. Also, make axes labels greater.

- Last sentence in caption of Figure 4: Is the stated conclusion reached by interpretation of the green dots? (By the way, you would probably get the same effect with an eccentric vertical axis). If yes, state it; if not, what is the utility of the green dots?

- Line 5 of section 4.2: After struggling myself to understand what do authors mean by this sentence (and an equivalent one in the Abstract), I think I finally got it. However, I think "leveling" applies to surfaces, rather than lines (the telescope line, in this case). I would suggest saying that "the method tolerates imperfect alignment of the telescope with the horizontal".

- Paragraph starting in line 13 of section 4.2: Avoid repeating what has been sated in item 2) above.

- First line of section 4.2.1: "**In areas of the globe with a small** geomagnetic inclination, **i.e., typically** within …"

---

## Referee Comment (RC2) · Anonymous Referee #2 · 5 Apr 2017

**1   General**

The article presents a novel method to perform and evaluate absolute measurements at geomagnetic observatories. This seems to be an useful generalisation of the traditional method that can provide easier identification of outliers, allows analysing more measurements points at once to improve accuracy of the measurements and can give the observer some more insight about sources of problems. These are all welcome benefits of the method, very useful to the community.

In the light of "allowing fellow scientists reproduction of results" I slightly miss some supplementary material (an appendix) with either source code or a fully worked out

example of calculations with some raw measurement data and steps to the final result that can be used to validate new implementations of the algorithm done by readers of the article.

The article heavily promotes the ability to do measurements without the zenith ocular. It would be nice to see that claim supported by some numerical results.

The abstract feels a bit long. The list of references might be shorter than usual, but since this is mainly an original mathematical article rather than a review article, this is probably fine.

The English language needs some polishing. The symbols in mathematical formulae could be introduced more smoothly (it is not always clear what they are without some background knowledge, sometimes they are simply explained two paragraphs "too late" to allow a smooth reading experience).

**2 General comments**

- I believe that it is relatively difficult for a newbie to fully understand the procedure and follow all the steps correctly. I would find it enormously helpful if you would include an Appendix with a fully worked out example (or potentially even example source code). Say, provide 16 measurement points (including data from variometer and $F$), the first eight being the usual ones (at vertical angles 90 and 270 degrees), followed by a bunch more more at arbitrary angles. Include some proper outlier and one typo that could be corrected (say, a reading that's 10 arc-minutes off) and compare the classical calculation with your method. I would find it important that someone who finds this article would be able to fully reproduce your results and verify that his/her implementation is 100% compatible with what you describe. While the article contains the formulas, they might feel a bit abstract, programming errors are easy to make and hard to debug.
- You claim that the method could be used at magnetic equator without the zenith ocular. While this is probably true, it might be interesting to see the comparison (either from a large set of simulated measurements, from mathematical analysis or from a measured dataset) between 8 points done using the traditional method (with the zenith ocular) and with another set where the four inclination measurements get replaced by a small number of measurements done at other angles. How does the uncertainty/error increase when we only measure at $90\pm60°$ or only at $90\pm20°$ ($90°$ for declination plus some more measurements at other angles). Plus the $270°$ counterparts of course.

- It takes quite some time to understand what Figure 1 is trying to tell. Personally I would find it useful to see a simple 3D plot of $F$ (with $D$ and $I$ angles marked somewhere), surrounded by a set of perpendicular arrows (perhaps also a formula) and a very short explanation saying that these perpendicular arrows represent all possible orientations of the magnetic sensor (and telescope) with $S = 0$. That is: orientations where we should be making our measurements. And explanation that the same (circle) is actually plotted in Figure 1 and 2.

  It would be nice to add (magnetic) latitudes of the three observatories next to Figure 1.

- Section 2.2 starts with a whole lot of equations, using symbols that have never been introduced before. Some are introduced after or with another equation. That makes the text more difficult to follow. You already use symbols $W \uparrow$ etc. in Figure 1 and don't even mention what that means. It would be more natural to me to introduce these symbols at the beginning of Section 2.1 where you introduce the traditional method and then continue using them throughout the article. In particular you never introduce $A_{W\uparrow}$ ($A_X$) (horizontal circle reading), $\Delta A_i$, ...

**3 Some notes about language**

Please note that I'm not a native speaker, so please double-check any language-related comment yourself.

But the English used should be improved. Many commas are wrong, many "a" or "the" are missing. I'm slightly puzzled by the "German constructs" like "allows to get", "allows to evaluate", "allows to incorporate", . . . . I think that "evaluating" or "evaluation" sound better.

The title of the article sound capitalised by the German rules. Shouldn't all words be either lowercase or all uppercase (other than prepositions)?

Theodolite **orientation** sounds better than **position** to me (both in title and later in text). You are not moving the theodolite around the room, you are only rotating it.

I would replace "$D$" with "declination $D$" at a few more places, in particular at the beginning. It's sometimes nicer to spell the words inclination and declination explicitly (not always of course).

**4 Minor issues**

**4.1 Abstract**

- Line 1: missing "**the** so-called"

- Line 2: "allows to get" sounds weird

- Line 8-9 (For absolute . . . ): The sentence might not be grammatically incorrect, but it is difficult to understand. I would use a different word order.
- Lines 11-12: perhaps "allows evaluation of an arbitrary number of measurements (...) at "arbitrary" theodolite orientations ...". The orientations are not really arbitrary ones. Five readings at $E_{\uparrow}$ would certainly not help. The article doesn't explain anywhere to what extend these measurements are in fact arbitrary.

- Line 15: **the** least square

- Line 16: "Gauss–Newton method" (no dash in front of method)

- Line 19: perhaps "The calculated residuals give/provide a measure of quality of each individual measurement"?

- Line 20: "allows to incorporate": sounds weird, maybe "additional measurements may be seamlessly incorporated"

- Line 27: What levelling errors are avoided? The user still needs to read the angle at least? (I also have a slightly hard time buying the argument that using just one adjustment wheel is that much easier.)

- Line 30: I would say "possible even with theodolites without **the** zenith ocular" ("the" missing also two lines lower)

**4.2 Introduction**

- Line 12: You say that you took advantage of methods from geophysical inversion theory. Is there some appropriate reference you could mention here?

**4.3 Section 2**

- please use $D$ and $I$ ($D$, $I$) rather than plain D and I

- The first paragraph sounds like an appropriate place to introduce symbols $W \uparrow$ etc. that are later used in Eq. 1 and later.

- Lines 19, 22: Do any theodolites have the sensor mounted below the telescope at $\xi = 90°$? Unless that's common, why not explicitly mention sensor up and down at this point and say "sensor below the telescope" rather than "sensor on the other side of the telescope"?

- Lines 20, 24: w**h**ere

- Lines 21, 24: I would say: "where the magnetometer reading $S$ is small or zero" and "where the field reading $S$ vanishes" (that is: include the symbol)

- Line 22: why do you use index $c$ for $\phi_{c1}$? Why not

$$\phi_1, \dots, \phi_4$$

(including comma)?

- Lines 23, 27: **the** magnetic North, **the** fluxgate, **the** telescope

- Shouldn't "eq. 1" be "Eq. 1", i.e. all uppercase, same for Figure $N$?

- Lines 31, 32: I would reword this.

- Figure 1 uses $\varphi$, while you refer to $\phi$ in the text. The same letter should be used.

- Symbols $W \uparrow$ etc. are already being used in the figure that should ideally be introduced earlier.

- I would make the $x$ and $y$ axis labels upright rather than italic.

- Figure description: I would say "The arrows indicate whether the sensor ..."

[Figure]

- The positions possible at a given observatory: "The sensor orientations at which one should measure at a given observatory should be . . . "

- Note the different $\phi$ values: this cannot be seen easily from the figures.

**4.4  Section 2.2**

- Line 4: I would say something like "Calculating declination $D$ and inclination $I$ from the conversional scheme is don by simply calculating mean values"

- I lack a gentler introduction to the variables and symbols being used in these equations. Some are not even defined/introduced or are introduced too late when the user has already lost focus and needs to come back.

- Page 4, line 3: Start the sentence with the names rather than parenthesis.

- Line 4: The sentence "In practice though ..." is suboptimal. We do repeated measurements all the time, they have to be done. You probably wanted to say that the entire measurement set is discarded along with the good readings?

- Line 5: **a** good reason

**4.4.1  Typography**

- Use `$\sin(x)$` and `$\cos(x)$` rather than `$sin(x)$` and `$cos(x)$` to get "$\sin(x)$" and "$\cos(x)$" rather than "$sin(x)$" and "$cos(x)$".

- Non-mathematical subscripts should not be typeset in math mode, use `\textnormal` to get $S_{\mathrm{reading},i}$ rather than $S_{reading,i}$. Same is true for "$apriori$", "$S_{off}$" (use $S_{\mathrm{off}}$), . . . (and blame the designer of this stylesheet for not matching the sizes of text and math to make even this look super ugly :)

- Degree sign in Eq. 1 and 2 should be in superscript.

**4.5   Section 3**

- Lines 9, 10: use symbols ($D$, $I$, $S$, $\delta$, $\epsilon$) next to descriptions

- Line 12: The general method described below allows evaluation of

- Line 13: even less **measurements** are possible

- Line 16: I would move "using the new method" at the end of the sentence

**4.6   Section 3.1**

- Line 20: allow calculating

- Eq. 3: I find it a bit confusing that you once use $f()$ with and once without $S_{\text{off}}$. But in any case I have troubles understanding how $c \cdot F \cdot f(\ldots, S_{\text{off}}) = \ldots + S_{\text{off}}$: what happens when $c = -1$ and why would $F$ influence the final $S_{\text{off}}$? I believe that $S_{\text{off}}$ should be strictly out of function $f$ from the start.

- Please look at typographical notes for previous equations and reduce the spacing after $F\cdot$.

- Page 5, line 1: I would put "$c = 1$ if $S > 0$ when the telescope is pointing towards North" in parenthesis

- Line 2: $S$ should be in math mode.

- Eq. 4: Shouldn't it be $\Delta D_i$ rather than just $\Delta D$ (same for $I$)?

- Line 12: what role does the word "respectively" play? (It's at a strange location anyway.)

- Eq. 6: you never defined $f_i$ nor $F_i$. What exactly is $f_i(\mathbf{p}, \phi, \xi)$ and why not perhaps $f(\mathbf{p}, \phi_i, \xi_i)$? How do you get from $f_i$ to partial derivatives of $F_i$ and what is $F_i$? You used $\Delta D_i$ here, but not in the second line of Eq. 4 and 5.

- Line 20: Ideally are ...: weird sentence. Perhaps "Ideally we would have $r_i = 0$ for each measurement $i$."

- Line 21: the number ... exceed**s**

- Line 22: **the** least square

- Eq. 7: the quantities used here are not explained or defined

- Page 6, line 2: **by** investigating

- Line 4: **the** first guess of $D$ and $I$ (use math mode)

- Line 6: "achieved" is not the right verb, "by" is missing

- Line 7: What exactly is a "3D pointing vector"? This reminds me of "Poynting vector" :)

- Use \cos and \sin

- Line 9: plan**e**

- Line 10: "$D$ and $I$ ar calculated accordingly.": it's difficult to explain what bothers me here, but I had a feeling I was supposed to know already how to do that after reaching the dot, but didn't know how. The next sentence explains how, but the flow of thoughts is broken and the last sentence is not even a standalone

sentence. You should probably at least change the dot after "accordingly" to a colon. Or do something else?

An image would help here. (It could be the same as I suggested adding to Figure 1.)

- Lines 11, 22, 24; page 7, lines 1, 2: capitalize "Equation"

- Line 15: **the** first guess

- Line 16: requires calculation of

- Line 17: known to have a quadratic convergence **which** is very fast

- Line 18: a or the? solution

- Line 20: Jacobian

- Line 21: with $N$ **being** the number (use math mode)

- Line 24: "according" doesn't sound like the right word

- Page 7, lines 1-3: we should elaborate on this earlier

- Line 5: magnetometer**s** (plural)

- Line 8: mechanic is a noun (person who repairs machinery), use "mechanical" instead

- Line 10: is often referred to as **a** variometer

- Lines 18-17: use math mode for all: $X$, $Z$, $Y$, $H$, $D$, $Z$, use `\arcsin`, use `\left(` and `\right` for parenthesis in Eq. 11, "var" and "abs" should not be in math mode

- Eq. 11: I would remove "or D" and simply write the "is approximately" as a single equation.

- Eq.13: For me it would be less confusing to see a single equation

$$Z_{\mathrm{abs}} = Z_B + Z_{\mathrm{var}} = F \cdot \sin(I)$$

- Line 21: identified -> associated? not sure

- Page 8, Figure 2: mark the "$x$" axis with $Y_{\mathrm{sensor}}$ on both graphs and the "$y$" axis on the left.

**5 Results**

- Line 4: the word "according" sounds strange to me at this place.

- Use math mode for $H$.

- Line 9: don't put units (nT) in math mode or better: use `\mathrm`

- Line 12: **it** is worthwhile

- Lines 13, 14: not sure if dates should be in British or American style, but this format is weird.

- Line 13: each measurement **done at?** 16 positions

- Line 14: According to our explanation the problem was in . . .

- Page 9, Figure 4: "is could be": remove "is"

- Page 10, line 2: allows making DI

- Line 10: "scheme method" sounds weird

- **the** declination measurement

- The verb **note** can also mean "pay attention to". This sometimes makes it a bit non-straightforward that one has to write the values down. At least for me.

- Line 16: use math mode for $D$

- Line 23: At Niemegk the inclination is $I = 67.5°$.

- Line 25 (also page 11, line 3): remove a set of Set :), put parenthesis around $90° - \alpha$ and add the forgotten degree sign (also missing in many other lines)

- fix capitalization of "Set" after commas in many lines

- Line 26: finish sentence and start "Adjust" with a capital.

- fix the rest of copy-paste errors

- Line 13: magnet**ic** equator

- Lines 16-17: math mode for $I$, Else -> Other, closed -> close, angel -> angle, bee -> be (no mythical creatures and bee hives in the article, please :)

**6 Discussion**

- Line 21: We have been testing . . . **at** the Niemegk . . .

- Line 23: data is usually singular

- Line 25: **the** major advantage Line 26: **on** the same day; allows improving, allows assessing

- Line 28: remove "a" from "a problems"

**GID**

Interactive comment

---

## Author Comment (AC1) · 14 Jun 2017

Dear reviewer,

I am really grateful for your numerous hints to unclear or mistakable wordings, verbal errors, typos and errors in equations! I tried to follow all your hints and advise. I hope this led to a substantial improvement of the paper.

I added a section, explaining how error bars are calculated and I corrected and completed the section about base line determination.

The abstract has been reduced.

Your major concerns were:

1) The instrument model is too simple and should include other parameters than misalignment errors and sensor offset.

   Replay: Of course, implementing the instrument model suggests including more than the three mentioned parameters. But that is not the scope of our paper. In the first place we wanted to generalize the set of used theodolite orientations. But yes, extending the data set to more orientations, distributed denser on the horizontal scale allows revealing effects of imperfectness not accounted for in the model. The most obvious one is certainly an instrument which is not leveled properly.

2) The advantage of the conventional scheme is, that the value of D is deduced from a set of 4 equations, while I is deduced from another set of 4 (virtually) independent equations.

   Replay: Is it really an advantage to separately solve two systems of four equations each? All information to be drawn out off differing results of both evaluations (D or instrument parameters) can also be seen investigating the residues of a common inversion. The result is even firmer determined, because only five instead of seven unknowns have to be calculated (seven because, I and delta are calculated twice). Accordingly, also the residues contain more

information. The first test we made, was of course applying the method to conventionally measured data. We got the same results as from the conventional scheme, but slightly firmer determined.

3) We mentioned now in the Introduction, that we to not aim to extend the simulated physics of a DI-theodolite. We only want to allow for the generalization using orientations off the traditional ones.

All your other hints to typos and errors were taken into concern. They have been corrected or led to a hopefully better formulation.

I hope I covered all your other concerns and apologize for the long time needed. Your contribution is really appreciated! It would be great if you could also contribute to the open discussion.

With best regards,

Heinz-Peter Brunke

---

## Author Comment (AC2) · 14 Jun 2017

Dear reviewer,

I am really grateful for your numerous hints to unclear or mistakable wordings, verbal errors, typos and errors in equations! I tried to follow all your hints and advise. I hope this led to a substantial improvement of the paper.

I added a section explaining how error bars are calculated and I corrected and completed the section about base line determination.

The abstract has been reduced.

Your major concerns were:

1) Missing source code.

   Replay: We will add the MATLAB source code as supplementary material to this publication. It still needs some revision and will be uploaded, possibly not before the interactive public discussion.

2) Evaluation of measurements without zenith ocular and example of outlier:

   Replay: I am going to provide simulated results as proposed.

3) It takes some time to understand what Figure 1 is telling.

   Replay: I tried to better explain the figure. But the matter is rather complicated. The plot allows nicely visualizing the set of orientations within a certain measuring scheme. We found out, that even people experienced with DI-measurements found it interesting to see, that all possible orientations are on two different lines and that you always need two rotations of 180° on both circles to get from a given sensor-up the respective sensor-down position. We thought about a 3D plot showing these things but could not find one. We added declination and inclination values of the three

observatories. They are the parameters determining the shape of the dashed lines specific to each observatory.

4) Use of symbols being incorrectly or to late introduced.

Replay: We tried to introduce now all symbols before using them.

Your effort to find so many errors in the text and in the equations is really appreciated!

I hope I covered all your concerns and apologize for the long time needed. Your contribution is really appreciated! It would be great if you could also contribute to the open discussion.

With best regards,

Heinz-Peter Brunke